# Assessing the Scale Effect on Bearing Capacity of Undrained Subsoil: Implications for Seismic Resilience of Shallow Foundations

**DOI:** 10.3390/ma16165631

**Published:** 2023-08-15

**Authors:** Zofia Zięba, Małgorzata Krokowska, Marek Wyjadłowski, Janusz Vitalis Kozubal, Tomasz Kania, Jakub Mońka

**Affiliations:** 1Department of Civil Engineering, Wrocław University of Environmental and Life Sciences, 50-375 Wrocław, Poland; zofia.zieba@upwr.edu.pl (Z.Z.); jakub.monka@upwr.edu.pl (J.M.); 2Faculty of Civil Engineering, Wrocław University of Science and Technology, 50-370 Wrocław, Poland; marek.wyjadlowski@pwr.edu.pl (M.W.); janusz.kozubal@pwr.edu.pl (J.V.K.)

**Keywords:** silty clay, scale effect, unconfined compressive strength, post-degradation shear strength, bearing capacity, seismic resilience

## Abstract

This research investigates the influence of the scale effect on the bearing capacity of fine-grained subsoil under undrained conditions. The analyses were conducted based on laboratory tests of silty clay. Uniformly compacted samples were subjected to an unconfined compression test. The research was performed on cylindrical specimens. Three different variants of the diameter D (38 mm, 70 mm, 100 mm) and the corresponding height H = 2D were analyzed. Based on the tests results, the unconfined compression strength q_u_ was determined, and from this, the undrained shear strength c_u_ was calculated. The obtained results showed a clear decrease in c_u_ with increasing sample size. However, in the existing reference documents, there are no specific guidelines for calculations of bearing capacity with consideration of sample size effect on the soil shear strength. Therefore, this study utilized the laboratory soil test data to calculate the bearing capacity of undrained subsoil, taking into account the seismic impacts, with a particular focus on spread foundations.

## 1. Introduction

Correct recognition of the physical and mechanical properties of the soil substrate enables safe and cost-effective design of engineering structures. In the case of weak soils, recognizing their geotechnical parameters allows for decisions to be made regarding methods of soil reinforcement. To accurately predict the behavior of soils under load, it is necessary to reflect the real conditions in which the soil substrate will operate. When determining the bearing capacity of the substrate, properly determining its strength parameters is crucial. If these parameters are determined under laboratory conditions, one of the factors that significantly influences this assessment is the size of the analyzed samples. The relationship between sample size and the measurement results of the mechanical properties of materials is referred to as the scale effect [1]. The scale effect is typically disregarded in engineering geotechnical analyses.

The analysis of the influence of the scale effect on soil strength has been the subject of research described in the scientific literature. In the early 20th century, it was assumed that the sample size should not reflect the characteristics of the tested material. In 1936, Parsons presented the results of laboratory tests contradicting this approach [2]. He first described the problem of the scale effect. The tests were conducted using a direct shear apparatus on samples of varying sizes formed from cohesionless soils. The results indicated that the larger the box of the apparatus (size of the tested sample), the smaller the internal friction angle. These findings were confirmed by subsequent studies [3,4], which also included the measurement of cohesion. In contrast to the internal friction angle, the value of cohesion increased with the increase in sample size. Other analyses involved the determination of transformation equations based on the results of laboratory tests. These linear equations allowed for the conversion of strength parameter values from small samples (60 × 60 mm) to large samples (300 × 300 mm) [5]. The phenomenon of the sample scale effect on obtained parameters is also evident in studies of composite materials. Researchers from Pakistan presented an analysis of the scale effect during direct shear tests on sand samples with different degrees of compaction using varying normal stresses and constant shear rates. The results confirmed previous considerations on the influence of the scale effect on the strength parameters of materials. In the same box size, the friction resistance increases with the increase in the volumetric density of the sand, while at the same volumetric density, this resistance decreases for larger sizes of the shear apparatus boxes [6].

Taking into account the effect of scale, it is possible to study it using a numerical approach with discrete spherical elements, which effectively represents loose noncohesive soils [7,8,9,10]. This method allows for the description of phenomena at microscopic scales of interactions, macroscopic scales of constitutive laws, and mesoscale objects, as demonstrated in various works. However, one limitation of this method is the difficulty in considering the liquid phase in the soil, particularly the existence of closed pores [7]. Additionally, the method’s reliance on the calibration of model parameters based on laboratory tests further poses challenges [8]. Due to these limitations, the method is not suitable for studying the effect of scale in cohesive soils. Some works have attempted to combine fluid flow through a skeleton of rigid spherical elements [11,12,13,14], but certain issues persist in both Discrete Element Method (DEM) and Finite Element Method (FEM) modeling including intermediate methods, like the finite difference method FLAC (Fast Lagrangian Analysis of Continua).

A promising modeling technique that overcomes the aforementioned problems is the hybrid FDEM (Finite-Discrete Element Method), which combines the principles of continuum mechanics with discrete element algorithms [15,16,17]. This approach proves valuable for modeling the interaction of many deformable bodies, including cracks. However, dealing with brittle fracture effects and scale issues still remains somewhat beyond the scope of the microscopic modeling presented in this work. Therefore, a good starting point for numerical studies would be an extended Drucker-Prager (D-P) model coupled with FDEM, which enables tracking pore pressures and recording cracks, along with encoding energy dissipation supplemented with Bazant correction factors.

Noteworthy research on the effect of scale in sandy soils filled with fine particles, relevant to the problem discussed in this paper, has been conducted by Skempton et al. [18] and Didier et al. [19]. These studies determined critical values for the initiation of flow and investigated the scale effect on these phenomena, analyzing convergence issues and indicating the existence of the scale effect. In light of the need for miniaturization to natural scale object models and ensuring result comparability, these works provide formulas to estimate the impact of the scale effect. Additionally, numerical calculations have been applied in soil mechanics to address liquefaction and earthquake effects for large-scale objects in various studies [20,21,22,23,24,25,26]. Meanwhile, solutions to fundamental modeling and numerical issues for earthquakes, particularly in FLAC 3D, have been explored through works analyzing previously recorded waveforms through model objects, with the scale represented by the thickness of the cohesive soil layer under the foundation [27].

The scale effect has been observed not only in soils, but also in many types of materials, such as metals [28,29,30,31], or in composite materials in terms of their strength properties [1,32,33,34]. Regarding the study of compressive strength, the scale effect is the subject of extensive research in the case of concrete samples. In the second half of the 20th century, various researchers investigated the dependencies between the compressive strength of concrete and the size of the samples used. The prevailing opinion was that larger cylindrical samples tend to result in lower compressive strength values, as determined by the ratio of the destructive force P to the sample surface area S [35,36,37]. The reason for this was that larger samples tend to have more inherent flaws, such as cracks or voids, which can weaken the overall structure and lower the measured compressive strength. Additionally, larger samples may be more prone to edge effects, where the compressive strength near the edges of the sample is lower than at the center. To account for these factors, various conversion factors were developed to adjust the compressive strength values obtained from different sample sizes. However, there is still ongoing research in this area, as the precise relationship between concrete sample size and compressive strength is complex and can vary depending on a range of factors, such as the type of concrete mix, the curing conditions, and the testing method used.

Granite, a rock characterized by low porosity, exhibits a distinct trend in uniaxial compressive strength (UCS) with respect to scale. This trend, as observed in various studies [38,39,40], follows a pattern of initial increase followed by a subsequent decrease. Conversely, certain rock types, like limestone, exhibit negligible scale effects on their mechanical properties. This phenomenon may be attributed to factors such as irregular structure, microcracks, and porosity, all of which intricately influence the relationship between scale and parameters like UCS and elastic modulus. Nevertheless, a comprehensive understanding and precise characterization of the internal irregular structure and its ramifications on rock mechanics remain elusive. Consequently, predicting the multifarious impact of scale on the mechanical behavior of weak porous materials remains a challenge. The influence of pore pressure distribution is paramount, particularly in materials like soils, where, unlike in rocks, an increase in pore pressure distribution leads to a reduction in the average UCS value. Another notable effect, contrary to the rock behavior, is the negative correlation between strength and sample size. This intricate interplay of factors underscores the complexity of rock mechanics in weak and porous materials.

In a recent study by Wu et al. [41], the investigation extended to 3D printed models of rocks illuminated the intriguing size effects present in their uniaxial compressive properties. Furthermore, efforts to encapsulate the in situ material behavior led to the formulation of a theoretical framework based on the concept of fracture energy [42]. According to this theory, the size effect in rock behavior stems from the dissipation of strain energy during the propagation of macroscopic cracks. Bažant’s pioneering work not only introduced the Size Effect Law (SEL), but also extended it to include fractal considerations, resulting in the Fractal Fracture Size Effect Law (FFSEL) [43]. These models account for the intricate fractal geometry of crack propagation and its implications for energy dissipation during expansion. The proposed models offer valuable insights into the size effect phenomenon, enabling more accurate predictions and analyses of rock strength variations across scales.

Research of the scale effects on the compressive strength results of soil samples has not been carried out as extensively as for concrete and rocks samples. Kamei and Tokida [44] performed the unconfined compression test (UCT) with different specimen sizes for reconstituted clays with different plasticity indexes. Their findings suggest that there is no significant effect of specimen size on strength and deformation characteristics when the ratio of specimen height to diameter is 2.0 for diameters greater than 20 mm. However, they observed a significant increase in q_u_ (unconfined compressive strength) and secant modulus (E_50_) values when the specimen diameter is reduced to 10 mm under the same height-to-diameter ratio of 2.0. This suggests that for smaller diameter specimens, the strength properties of the clay may be affected by the specimen size. It is important to note that these findings apply specifically to reconstituted clays and may not necessarily be applicable to other materials or testing conditions. Shogaki [45] has also reported that there is no significant effect of specimen size on strength properties when the height-to-diameter ratio is approximately 2.0. However, he mentioned that there may be variations in the results depending on the type of soil and the range of strength and plasticity. To determine the effect of specimen size on strength properties for a wide range of soils, systematic studies would be required.

The aim of this study is to analyze the influence of the scale effect on the strength of soil under unidirectional stress conditions during undrained shear (c_u_), as well as to investigate the impact of this parameter on the bearing capacity of shallow foundations during seismic actions. Additionally, an analysis was conducted to minimize the effects of the scale effect and estimate the level of safety depending on the applied standardized measurement method.

## 2. Materials and Methods

The schematic diagram shown in Figure 1 illustrates the research process and data analysis, along with their potential application for determining the bearing capacity of the substrate.

In the following subsections, a description of the materials used in the research, laboratory testing methods, and methods for calculating the bearing capacity of the soil under undrained conditions are presented.

### 2.1. Characteristics of the Tested Soil

The research was conducted on samples of fine-grained soil taken from the deposit of clay raw materials for building ceramics located southeast of the town of Bolków in the Lower Silesian Voivodeship, Poland. Initially, the content of individual soil fractions was determined [46]. Based on this, it was determined that the tested soil is a silty clay (Figure 2).

Moreover, Atterberg limits (LL, PL) [47], maximum dry density of the soil skeleton (MDD), and its corresponding optimum moisture content (OMC) [48] were determined. The filtration coefficient (k) [49] and the specific surface area influencing interparticle interactions (S_0_) [50] were also calculated. The properties of the analyzed soil are presented in Table 1.

Additionally, a general analysis of the soil’s microstructure and chemical composition was conducted using a scanning electron microscope (SEM) (Figure 3).

Based on this, it was observed that the particles of the analyzed soil are sub-angular and rough, predominantly composed of silicates and aluminosilicates.

### 2.2. Laboratory Tests

The undrained shear strength c_u_ was determined in accordance with the technical specification [51]. This method is widely used due to its low cost and time efficiency [52,53,54,55]. These tests provide comparable results to the unconsolidated undrained (UU) test conducted in a triaxial compression apparatus [56]. Selecting this method is appropriate due to the very low permeability of the tested soil, represented by the coefficient k specified in Table 1. The primary aim of the laboratory tests was to eliminate factors other than the scale effect that could influence the strength of the samples. Therefore, the tests were conducted on uniformly compacted samples. The prepared soil with optimum moisture content (OMC) was compacted manually to the maximum dry density (MDD) in a given volume (Table 1). Each time, the samples were weighed and measured to control compaction, as well as the moisture content was determined after the test. In all groups, the maximum measurement error did not exceed 1%. This ensured that regardless of the sample size, all specimens had the same compaction and moisture content. Additionally, the tests were carried out in a room with a controlled and constant temperature.

A total of 45 cylindrical, uniformly compacted samples with varying diameters D were subjected to unconfined compression tests (UCSs), with 15 samples in each group hereinafter designated as Group 1 (D = 38 mm), Group 2 (D = 70 mm), and Group 3 (D = 100 mm). The height-to-diameter ratio was maintained at H/D = 2. The calculation of the bearing capacity of a foundation subjected to horizontal seismic primary (P) waves (compressional or longitudinal waves that push and pull the ground in the direction the wave is traveling) requires the application of the strength concept with the parameter c_u_ (1). The undrained shear strength c_u_ was taken as half of the compressive strength under uniaxial stress q_u_ (1), which is defined as the maximum axial stress at failure (2) [51].
c_u_ = 0.5 × q_u_,(1)
q_u_ = σ_v_ = P/(A_i_/(1 − ε_v_)),(2)
where c_u_—undrained shear strength [kPa], q_u_—unconfined compression strength [kPa], σ_v_—vertical stress [kPa], P—vertical load [kN], ε_v_—vertical strain [−], A_i_—initial cross-section area of the sample [m^2^].

### 2.3. Calculations of Bearing Capacity of Spread Foundation 

Earthquakes are natural phenomena that occur when Earth’s crust is subjected to stress and strain. The stress accumulates over time due to the relative motion of tectonic plates, which are large segments of Earth’s lithosphere that move over the underlying asthenosphere. When the stress exceeds the strength of the rocks along a fault, which is a fracture or zone of weakness in the crust, a sudden rupture occurs, releasing a large amount of energy. This energy propagates in the form of seismic waves, which are vibrations that travel through Earth or along its surface. Seismic waves can be classified into two main types: body waves and surface waves. Body waves can travel through the interior of Earth and are divided into primary (P) waves and secondary (S) waves. P waves are compressional waves that cause the particles of the medium to vibrate in the same direction as the wave propagation. S waves are shear waves that cause the particles of the medium to vibrate perpendicular to the wave propagation. Surface waves can travel only along the boundary between different layers of Earth, such as the crust and the atmosphere. Surface waves are divided into Love waves and Rayleigh waves. Love waves are horizontally polarized shear waves that cause the ground to move side to side. Rayleigh waves are elliptically polarized waves that cause the ground to move up and down and side to side in a rolling motion [57,58,59].

In this subsection, the methodology for calculating the bearing capacity of shallow foundations on soil subjected to seismic actions is presented. Considering the influence of seismic effects on buildings and structures requires recording ground acceleration values and creating a seismic action database for a specific location. Inertial forces resulting from seismic effects are horizontal actions exerted on buildings and structures. Dynamic loads are applied to all material points of the dynamic system or to a designated part of the model representing the foundation [60,61].

The bearing capacity of the soil under seismic actions is evaluated under undrained conditions. Under undrained conditions, the design bearing capacity of the soil can be determined following the procedure provided in the European standard Eurocode 7 [62], Formulas (3)–(6).
R/A′ = (π/2 + 2) × c_u_ × s_c_ × i_c_ × b_c_ + q [kPa],(3)
where R—resistant value [kN]; A′ = B′ × L′—effective area of the foundation [m^2^]; q—external load [kPa]; s_c,_ i_c_, b_c_—dimensionless coefficients considering the influence of:shape of the foundation base (4)
s_c_ = 1 + 0.2 × B′/L′ [−](4)

inclination of the load induced by horizontal load H [kN] (5) i_c_ = 0.5 × (1 + (1 − H/(A′ × c_u_))^0.5^) [−](5) with the caveat that H ≤ A′ × c_u_.

slope of the foundation base (6) b_c_ = 1 − (2 × β)/(π + 2) [−](6) where beta is the angle of inclination of the foundation base in radians.

Horizontal actions always have an unfavorable effect on the bearing capacity of the soil. In the computational model, this fact is taken into account by the coefficient i_c_, which takes a value of i_c_ = 1.0 for a scheme without horizontal force and a value of i_c_ < 1.0 for a system of actions where horizontal force H is present.

The exploration of fine-grained soils in Poland (Figure 2), and their juxtaposition with materials from diverse global regions within the realms of geotechnical engineering and soil mechanics, illuminates a recurring theme of discerning mechanical and physical attributes. Primarily, this endeavor revolves around deriving values that seamlessly transition into subsequent design phases. It is imperative, however, to acknowledge the substantial influence of geological factors, with certain material traits intrinsically linked to specific locales. Although these properties can exhibit substantial variations across diverse regions due to geological, climatic, and environmental distinctions, their underlying descriptive framework remains consistent. As a result, the findings presented in these studies universally pertain to the characterization of fine-grained cohesive soils with attributes such as low porosity and optimal compaction. This universality highlights the broader relevance of the issue, extending beyond Poland to encompass seismic regions worldwide.

Failures of wooden, masonry, or mixed construction buildings and structures subjected to seismic actions occur as a result of exceeding the load-bearing capacity of structural elements [63]. The development of reinforced concrete structures and the skillful shaping of their resistance to seismic influences [64] have revealed failures that stem from the loss of soil bearing capacity while maintaining the integrity of the structure. Toppled and leaning high-rise buildings after the magnitude 6.0 earthquake struck Hualien (Taiwan) are an example of this phenomenon (Figure 4).

Adopting the value of the undrained shear strength parameter (c_u_) as the fundamental geotechnical parameter of the soil requires laboratory testing, which is presented in the study while considering the effect of sample scale. The bearing capacity of the soil is also dependent on the value of the horizontal force acting on the foundation, which results from permanent and variable loads, including exceptional seismic actions. The values of seismic actions are determined based on historically observed accelerograms.

As a case study of the influence of the laboratory sample scale effect on the bearing capacity of the soil, calculations are presented for a prefabricated foundation with dimensions B × L = 2.0 × 3.0 m^2^, founded on cohesive soils. This type of foundation represents a standard footing for power poles. The seismic conditions occurring in the Legnica-Głogów Copper District (LGOM) in southwestern Poland were adopted. The computational values of seismic ground accelerations were based on the zones described in the reference literature [66] indicated below.

Zone where the expected maximum ground velocities (v_g_) are less than 1 cm/s, and no measures to counteract mining-induced vibrations are planned.Zone where the expected maximum ground velocities are 2 cm/s > v_g_ > 1 cm/s, and only limited countermeasures are planned.Zone where the expected maximum ground velocities are 4 cm/s > v_g_ > 2 cm/s, with a = 40 cm/s^2^.Zone where the expected maximum ground velocities are 6 cm/s > v_g_ > 2 cm/s, with a = 60 cm/s^2^.

In light of the increasing resilience of above-ground structures, the development of secure foundations against seismic impacts remains a critical facet of geotechnical engineering. By integrating the comprehensive analysis of soil properties and their dynamic interaction with seismic effects, the methodology outlined herein facilitates the creation of foundations that withstand the challenges of varying terrains and seismic conditions.

## 3. Results

Preliminary analyses revealed that upon exceeding the ultimate stress state, the specimens experienced visible deformations. Regardless of their size, the most frequently observed type of failure (approximately 95% of the samples) was characterized as “brittle” failure, with a distinct shear plane clearly visible (Figure 5).

This plane runs at a similar angle for groups with diameters of 38 mm, 70 mm, and 100 mm. This serves as a kind of confirmation of the homogeneity of the compaction of individual samples, which affects the strength parameters of the soil [67]. Additionally, to illustrate the nature of sample failure, the results of uniaxial compression are presented in the form of the vertical stress versus vertical strain relationship for a set of individual samples (Figure 6).

Preliminary findings indicate that the maximum stresses at failure, assumed as the compressive strength under uniaxial stress conditions, decrease with increasing sample size while simultaneously increasing the vertical deformation.

Detailed results of the compressive strength under uniaxial stress conditions (q_u_) and the shear strength without drainage (c_u_), calculated for individual variations of soil sample sizes (Equations (1) and (2)), are presented in Table 2. These results represent a complete set of tests for a group of three sample diameters (Ds) and volume (V).

For each sample size variant, a statistical analysis was conducted on the obtained undrained shear strength results. The arithmetic mean and the standard deviation SD were calculated, and the minimum and maximum values were determined, as well as the lower and upper confidence limits of the standard deviation at a confidence level of 95% and 99%. To assess the dispersion of the results, the coefficient of variation CV was determined (Table 3). It is generally accepted that if the value of this parameter is less than 10%, the data set within the groups is consistent [68].

In the case of samples with a diameter of D = 100 mm (Group 3), one outlier value was rejected, while for samples with a diameter of 100 mm, two values outside the range of the double standard deviation (SD) from the mean were discarded. Based on the average values of the remaining results, a graphical representation was prepared (Figure 7).

The ranges of values for individual groups, median, quartiles, and interquartile ranges can be compared in Figure 7. Comparing medians using graphical representation through a box plot corresponds to one-way analysis of variance (ANOVA) *t*-tests between groups. The data are also presented in the form of quantiles in Figure 8, which allows tracking the variability ranges of c_u_ for individual sample scales. For the sample with a diameter of D = 38 mm, all quantiles of c_u_ are outside the ranges obtained for samples with D = 78 mm and D = 100 mm. For the remaining two samples, the variability is high for quantiles below 0.9, while the upper limit overlaps.

These observations are also confirmed by the histograms created for each group and presented in Figure 9.

Based on the presented analysis, it was observed that both compressive and shear strength decrease with increasing sample size. However, these changes are not linear, but exhibit an asymptotic trend towards a value close to the mean for the scale D = 100 mm. The failure mechanism for the tested cohesive soil and soft rocks at the mesoscale and macroscale can be described as a material that undergoes plastic-brittle cracking, thus exhibiting a scale effect. The study identified a strength associated with the plastic failure mechanism, which is represented by a horizontal asymptote for the natural scale (here understood as D >> 100 mm). Therefore, it was assumed that there exists a shear strength value below the region of brittle fracture occurrence, referred to as the post-degradation shear strength c_upl_.

The exponential model approximating the relationship between the sample size and the value of c_u_ is described by Equation (7).
c_u_(V) *=* c_upl_ − (c_upl_ − c_0_) exp(−m V)(7)
where V—volume of the sample in liters; c_upl_—post-degradation shear strength; and c_0_, m are fitting parameters.

The asymptotic regression model functions in the R program were fitted using automatic parameter initialization in the NLS package, referred to as asymReg. The fitting parameters obtained for the data from Table 2 are as follows in Table 4.

Figure 10 graphically shows the results of the exponential curve fitting for the results from the three experimental groups. The figure also illustrates the confidence interval (for a level of 95% alpha = 0.05) for the asymptote corresponding to c_upl_.

Since the three parameters of the model were optimized, their analysis related to confidence levels is presented graphically in Figure 11 as the profiles of t-statistics. They are defined as the square root of change in sum-of-squares divided by the residual standard error with an appropriate sign (modulus for c_upl_, m, and c_0_ has been used), and presented also in Table 5.

In this study, the sample volume is utilized as a reference dimension, enabling the visualization of the relationships between various sample proportions and the maximum dissipation of destructive energy, while maintaining a constant slenderness ratio across all tested groups. Additionally, it is postulated that beyond a certain sample size, a reduction coefficient Kr (Equations (9) and (10)) can be determined to mitigate the scale effect and obtain the residual shear strength of the soils, referred to as the post-degradation c_upl_ in this context. It is also hypothesized that the increase in material strength is constrained by the minimum sample size applicable within the mesoscale range. The asymptotic value of the proposed model for larger volumes is defined as the desired and reference value for the c_upl_ material (8).
c_upl_ = (c_u_ exp(m V) − c_0_)/(exp(m V) − 1)for exp(m V) ≠ 1for c_u_ ≥ c_0_/exp(m V)(8)
where the reduction coefficient is defined as dependent on the fitting parameters and sample volume (9).
K_r_ = c_upl_/c_u_.(9)

Hence, after necessary transformations, the value of the coefficient is described by Equation (10).
K_r_(c_u_, V) = (c_u_ exp(m V) − c_0_)/(c_u_ (exp(m V) − 1)).(10)

The values of c_0_, m, and their functions are constants, thus the distribution of the c_upl_ parameters can be derived from the initial distribution of c_u_. The first central moment has a value dependent on E[c_u_], described in Equation (11)
E[c_upl_] = (exp(m V) − 1)^−1^(E[c_u_]exp(m V) − c_0_),(11)
and after substituting the values, the second central moment can be obtained using Formula (12).
Var[c_upl_] = (E[c_u_^2^] − E[c_u_])^2^) (exp(m V)/(exp(m V) − 1))^2^(12)

Based on Table 6 showing the values for the point estimate of distributions where the input parameter is the assumed distribution c_u_ and the result is the estimation of the parameter c_upl_, this allows estimation on the basis of the study of a selected set of samples with their known statistical description to determine the sought description of the design parameter. The proposed formula for the ultimate load capacity of the foundation after minimizing the effect of scale (for determining the load capacity in the case of compression testing of the material under no-drain conditions) is described by Equation (13).
R/A′ = (π/2 + 2) × c_upl_ × s_c_ × i_c_ × b_c_ + q [kPa].(13)

In the included task, the impact of the earthquake is taken into account as a horizontal force transmitted to the foundation (Figure 12). The value of the horizontal force depends on the expected acceleration of the ground. In order to emphasize this influence, the calculation scheme (Figure 13) does not provide for eccentric or moment loading.

The ultimate load bearing capacities of a typical direct foundation for power poles for different values of the c_u_ parameter and seismic acceleration are presented in Table 7. The bearing capacity of the subsoil is shown for varied cases of impacts: static and dynamic impacts of varying intensity. The estimated bearing capacity of the subsoil differs significantly (about 50%) depending on the assumed mean value of soil parameters obtained for three groups of samples.

The bearing capacity of the foundation is described by Equations (3)–(6); it was coded in S language (R program) as a multivariate function based on Equation (12).

After 10^6^ simulations of the value of the nonlinear bearing capacity Formulas (3)–(6), where the random variable for a fixed volume of the sample was the value of the experimentally measured c_u_ value (consistent as to point estimate and dispersion with respect to the mean), the results are presented in Table 8 and in the graphical form in Figure 14. Results are based on the assumption that the bearing capacity is calculated directly after laboratory tests of samples from selected groups. 

Figure 14 shows the different-than-expected scatter of the results. The smallest SD value is associated with the population parameters of the Group 2 samples, the samples of Group 3 rank in an intermediate place, while the SD result for Group 1 is as expected and gives the largest uncertainty range of the results.

## 4. Discussion

In practice, the most commonly identified factors influencing the value of c_u_ are the heterogeneity of the soil material in terms of its grain size, compaction, and moisture content. Climate conditions, such as temperature and air humidity, which can change the soil parameters over time, especially in the near-surface layer, are also significant. Considering that the analyzed samples were prepared with great care to maintain isotropy and eliminate any heterogeneity of soil parameters, and the fact that the testing conditions in the conducted experiment were constant and controlled, the aforementioned factors did not affect the obtained results. The dimensions of the samples in the series have a significant impact on the strength values. The samples were made of high-quality brass molds with high stiffness, hence the dimensional differences in the series are negligible. The loading was applied using a strength testing machine with guides that prevent off-axis and non-parallel loading. Therefore, there are no imperfections in sample dimensions or loading application. In all tests, the contact surface was prepared using the same technique and with the same surface roughness of the press material.

Looking for the source of the significant differences in the strength values of samples of the same proportions, but different volumes (diameters), the authors find the potential origin in the seemingly universal measurement methodology. Compression at a constant rate of strain was the source of inhomogeneities, as the material is homogenic; however, the small amount of moisture (semi-saturated conditions) inside pores is the source of relaxation internal pressure. In the experiments, the apparatus did not measure pore pressure (air and water) on the contact plates and inside material. However, as we deduced after eliminating other sources of inhomogeneities, the different way of dissipation of energy of both mediums (air and water) inside samples from different groups was a possible source of lower values of c_u_ in Groups 2 and 3 rather than Group 1. However, the type of solution of the Richards equation [69] suggested a nonlinear form of pressure profile in this same time of deformation of the samples with groups. Despite the diagnosis of this inhomogeneity, the method indicated in the paper assigning scaling factors to the material allows at low cost (without lost internal pressure sensors) and using standard test tools (sample volumes) to determine the values of c_upl_. At the same time, the demonstrated existence of c_upl_ different in value from the measured c_u_ in the apparatus with direct uniaxial testing should have consequences in the emergence of new values of safety factors for foundations designed on the basis of c_u_.

Based on this, it can be inferred that a higher compression rate for larger samples could have caused higher pore water pressure, resulting in higher stress concentration on their surfaces. Based on this assumption, the failure process for taller samples could have been initiated at lower average stress values on the surface. Can the scale effect be reduced by reducing the loading rate? This should be the goal of further research considering pore pressure and the adaptive nature of sample loading.

The results are concentrated in three volume groups, which makes it difficult to fit other approximating models, such as the logistic curve with more parameters. The applied model is a three-parameter concave-down exponential curve that decreases throughout the entire domain and does not have an inflection point, but it has a horizontal asymptote. The fitting function ensures the continuity of the curve and its derivative, and the fit is very good, especially for the nonlinear distance between the investigated sample groups measured by their volumes. The asymptotic values of c_upl_ show narrow ranges of variability with high confidence coefficient values. The fitting is resistant to measurement uncertainties, where attention is drawn to high coefficients of variation for two measurement groups. By using the exponential curve, it is also possible to predict the material strength for samples with sizes beyond the range covered by the experiments, allowing for a wider utilization of strength data in structural design.

In the literature [5], the proportion between a selected sample dimension and the D_50_ from the grain size curve is used to describe scales. In the examined cases, considering this proportion for a constant value of D_50_ = 0.011 mm is applicable and provided in Table 9 for the mean values in the groups. However, it inherits both the advantages and disadvantages of using only the D dimension. Which scale description should be chosen? The authors, due to the constant height-to-width ratio, chose volume as the measure of energy dispersion. Whether this is the optimal choice should show in the next program of research eliminating the doubts indicated earlier.

## 5. Conclusions

The scale effect is a phenomenon observed in many fields that rely on experimental measurements. Determining the c_upl_ values through uniaxial compression requires conducting a series of tests for different sample slenderness ratios. The boundary state surface described by the Tresca hypothesis does not consider the effects of brittle failure and is therefore not sensitive to the scale effect. Proper selection of strength parameters of geotechnical materials based on the available and applied measurement database is a cognitive problem that requires multiscale research. An important aspect of the conducted work is the use of typical measurement sets in geotechnical laboratories focused on small samples. This approach is reasonable for fine-grained soils, but the scale effect leads to significant overestimation of the material’s mechanical parameters and simultaneously the load capacity of shallow fundament. It is necessary to introduce a reduction factor, and the proposed solution allows for recalculating values based on small-sized samples to results similar to in situ tests. A very high coefficient of variation (and standard deviation) was obtained for the experiments in Group 1, and the authors do not recommend these tests for c_upl_ analysis. The study analyzed the impact of the scale effect on shear strength without drainage in uniaxial stress conditions, which is particularly important from the perspective of calculating the bearing capacity of soil. Based on the conducted research, the following conclusions were drawn:The sample size has a significant influence on the results of shear strength without drainage in uniaxial stress conditions. In the analyzed compacted silty clay, the shear strength (c_u_) decreases with increasing sample size. This corresponds to a difference of 31.74% between the strength of a sample with a diameter of 38 mm and a diameter of 100 mm, 22.90% between a sample with a diameter of 38 mm and a diameter of 70 mm, and 11.46% between a sample with a diameter of 100 mm and a diameter of 70 mm.The scale effect should be considered in geotechnical design because the results of tests on samples with a diameter of 38 mm, which are most commonly performed, show overestimated values. Consequently, in engineering practice, this can lead to damage or even disasters during the operation of structures.The conducted research confirms the significant influence of the scale effect on the strength of soils in uniaxial stress conditions. Smaller samples exhibit higher strength, which confirms the results of previous studies [4,70]. The phenomena of scale effect for direct shear box tests and footing bearing capacity were studied for noncohesive soils. This work extends the scope of research to cohesive soils and uses more advanced measuring devices.The scale effect distorts the results of laboratory tests, and a reducing coefficient should be established in relation to the actual dimensions of the considered medium [6]. For all samples, the average value of the transformed results is similar to the value obtained from previous experiments. Significant differences arise in terms of variance. The use of samples with small dimensions (Table 6, Group 1) to determine c_upl_ values results in a considerable uncertainty burden. The standard deviation of the transformed result increases several times. Samples with larger dimensions (Table 6, Groups 2 and 3) show no significant differences.The tests of materials under undrained conditions (similar to those analyzed in the study) require using samples with minimum dimensions of Φ = 70 mm and a height of 140 mm or larger.The boundary task of the bearing capacity of the direct foundation subjected to additional earthquake loads, by analyzing the dispersion of the results in relation to the average bearing capacity, supports the argument of the need to determine the value of c_upl_ with the smallest margin of error, as it carries over to the results, multiplying the uncertainties in the estimation of the bearing capacity. Samples with intermediate dimensions provide the most reliable estimation of foundation bearing capacity, with the least dispersion from the c_upl_ mean value.

## Figures and Tables

**Figure 1 materials-16-05631-f001:**
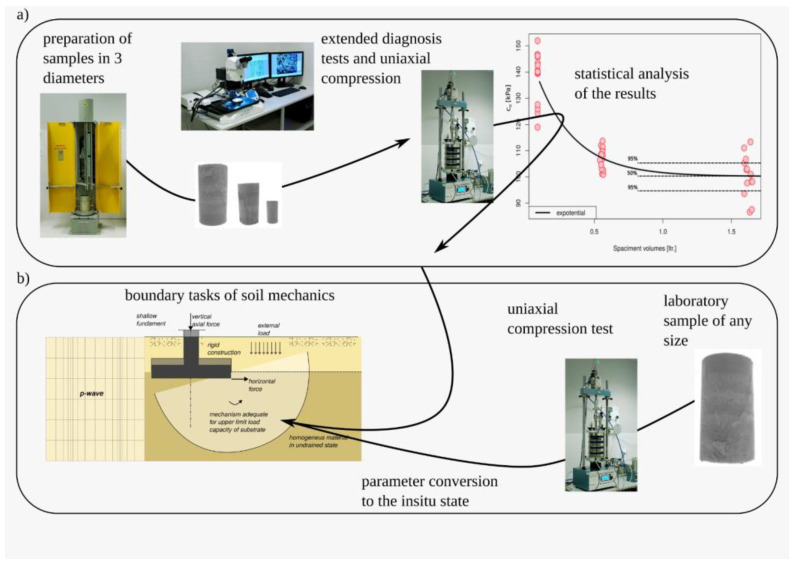
Diagram presenting the primary propositions of the article, delineating the research component (**a**) and its applicability in load capacity estimation (**b**).

**Figure 2 materials-16-05631-f002:**
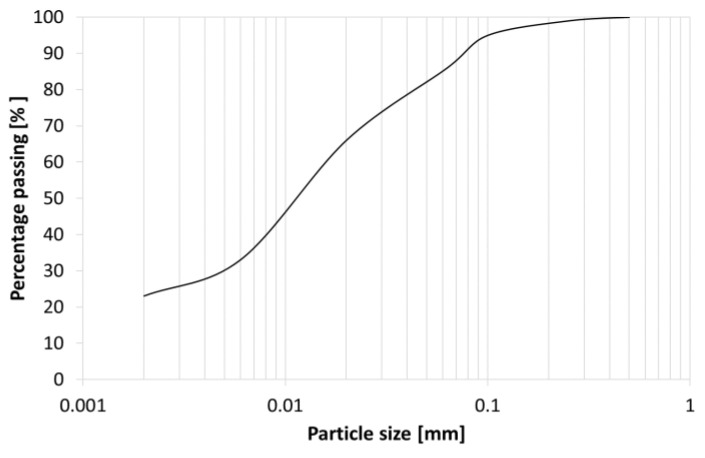
Grain size distribution of tested soil.

**Figure 3 materials-16-05631-f003:**
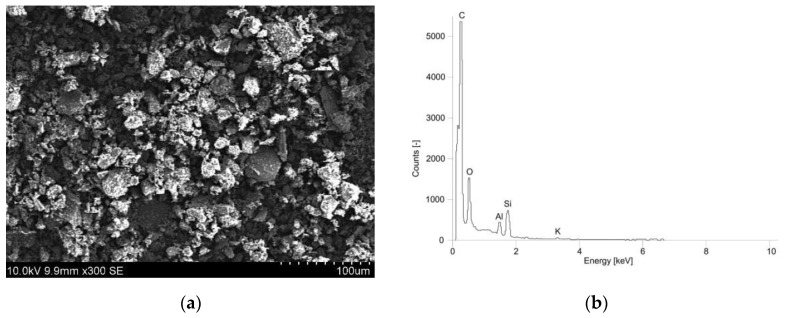
Soil microstructure (**a**) and EDS spectrum (**b**) obtained from the surface based on SEM image.

**Figure 4 materials-16-05631-f004:**
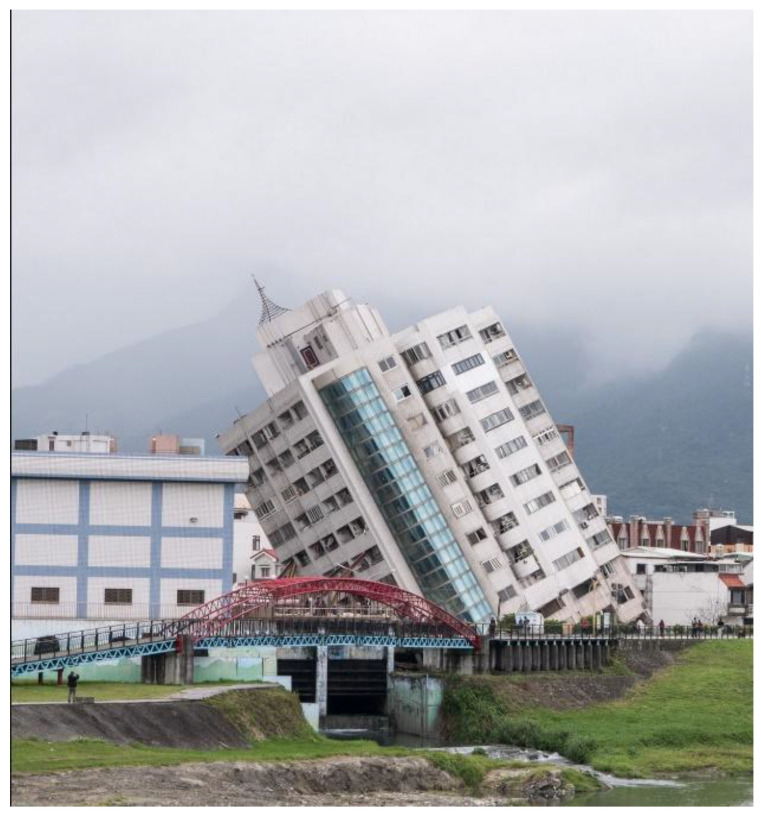
Collapsed residential building after reaching the ultimate limit state of the subsoil, Taiwan 2018 (photo: Cho Hsun Lu, CC BY 3.0) [65].

**Figure 5 materials-16-05631-f005:**
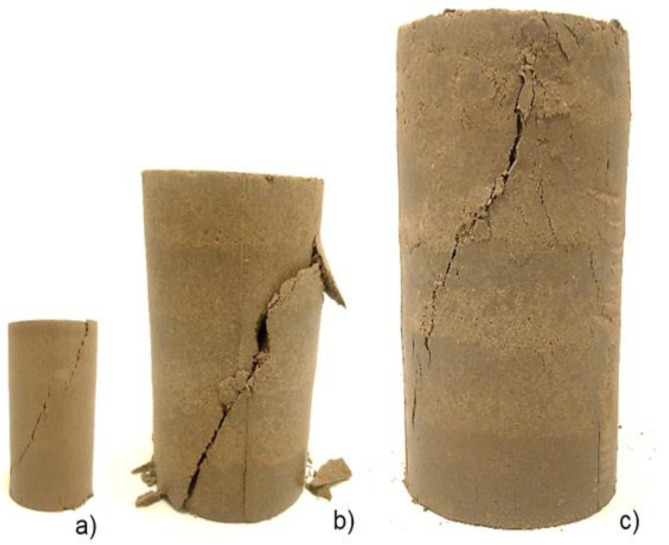
Most common type of sample failure in UCS tests: (**a**) Group 1—38 mm; (**b**) Group 2—70 mm; (**c**) Group 3—100 mm.

**Figure 6 materials-16-05631-f006:**
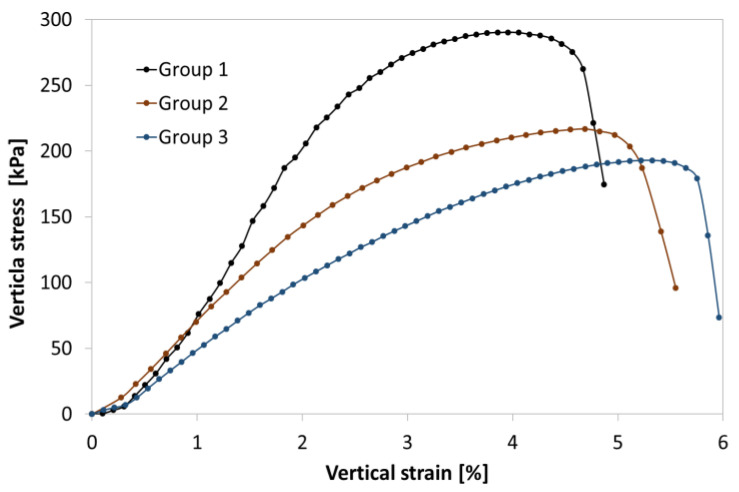
Stress–strain curves of tested groups of samples of compacted silty clay under unconfined compression: Group 1—38 mm; Group 2—70 mm; Group 3—100 mm.

**Figure 7 materials-16-05631-f007:**
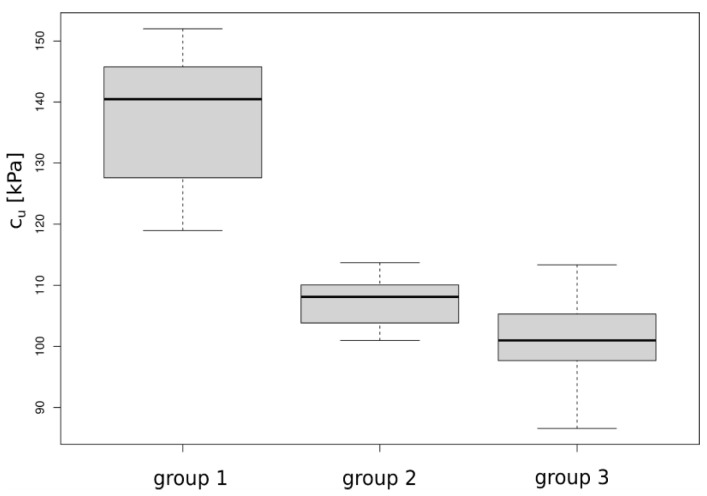
Basic statistical description for the tested groups in the form of box plots visualizing the Minimum Non-Outlier, First Quartile, Median, Third Quartile, and Maximum Non-Outlier.

**Figure 8 materials-16-05631-f008:**
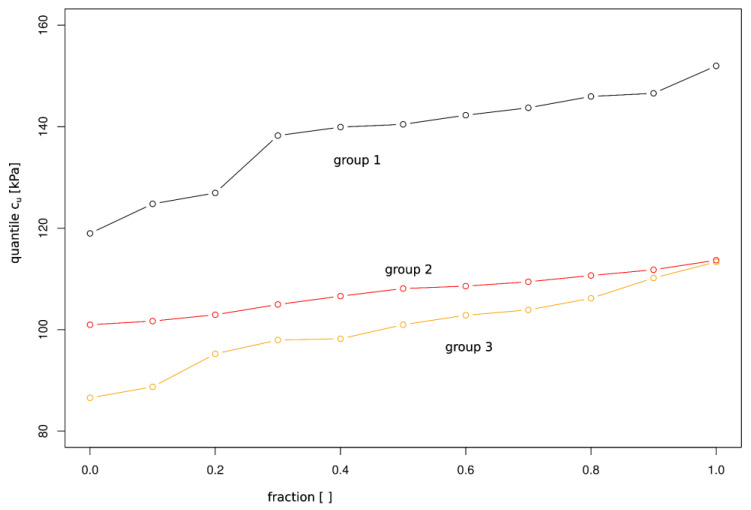
Shear strength statistical analysis. Quantile comparison across three tested groups of compacted silty clay: Group 1—38 mm; Group 2—70 mm; Group 3—100 mm.

**Figure 9 materials-16-05631-f009:**
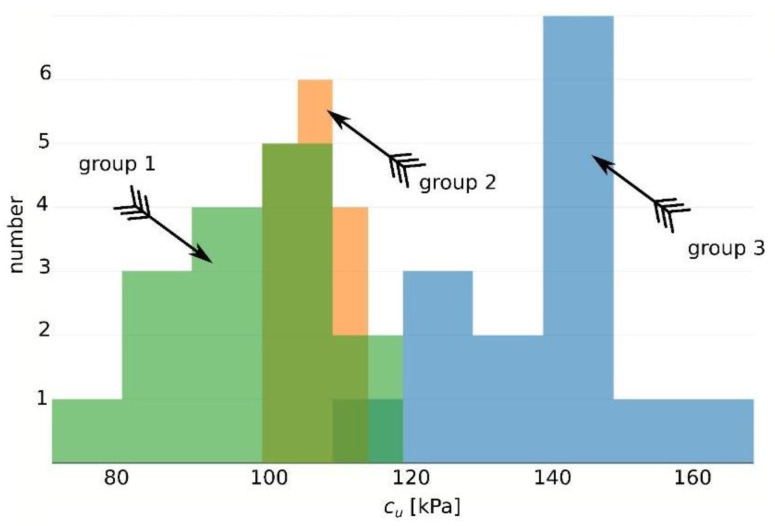
Distribution of results of shear strength c_u_ in tested silty clay groups: Histograms.

**Figure 10 materials-16-05631-f010:**
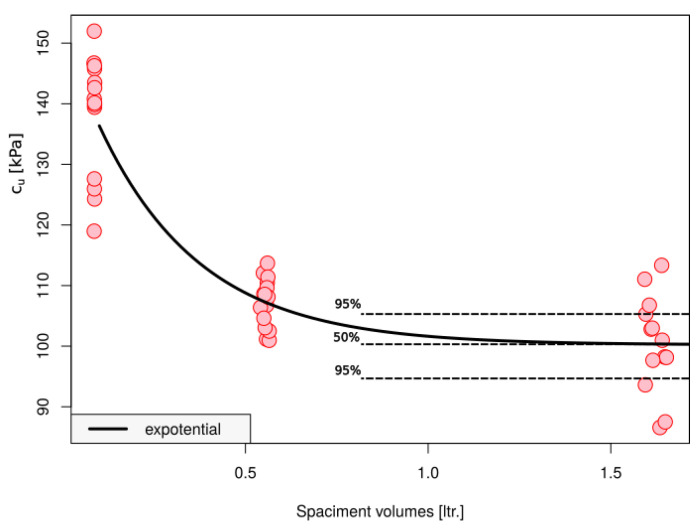
Fit exponential function to data. The lower and upper confidence limit for c_upl_ is presented as a gray field for alpha = 0.05.

**Figure 11 materials-16-05631-f011:**
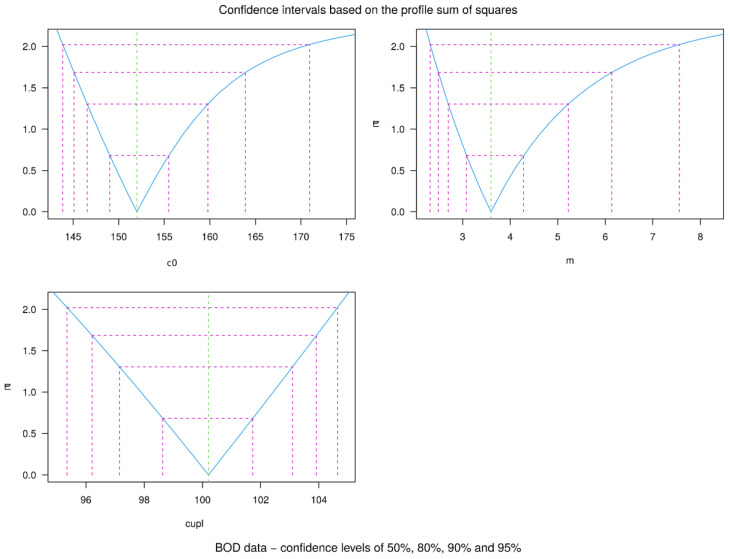
Confidence intervals based on the profile sum of squares for c_upl_, m, and c_0_.

**Figure 12 materials-16-05631-f012:**
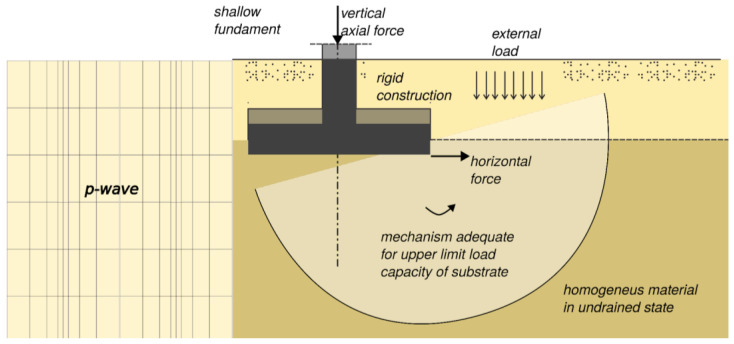
Mechanism for shallow fundament with condition related to presented task (the depth of fundament is limited to L′) in face to p-wave.

**Figure 13 materials-16-05631-f013:**
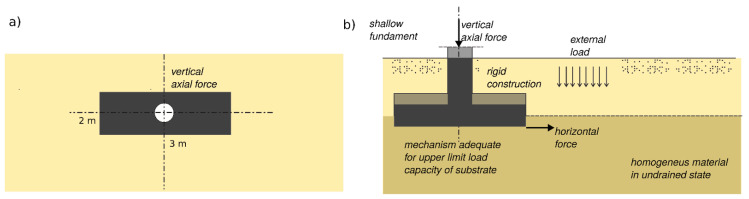
The description of the analyzed problem: (**a**) dimensions and shape (view), (**b**) geotechnical conditions and loading.

**Figure 14 materials-16-05631-f014:**
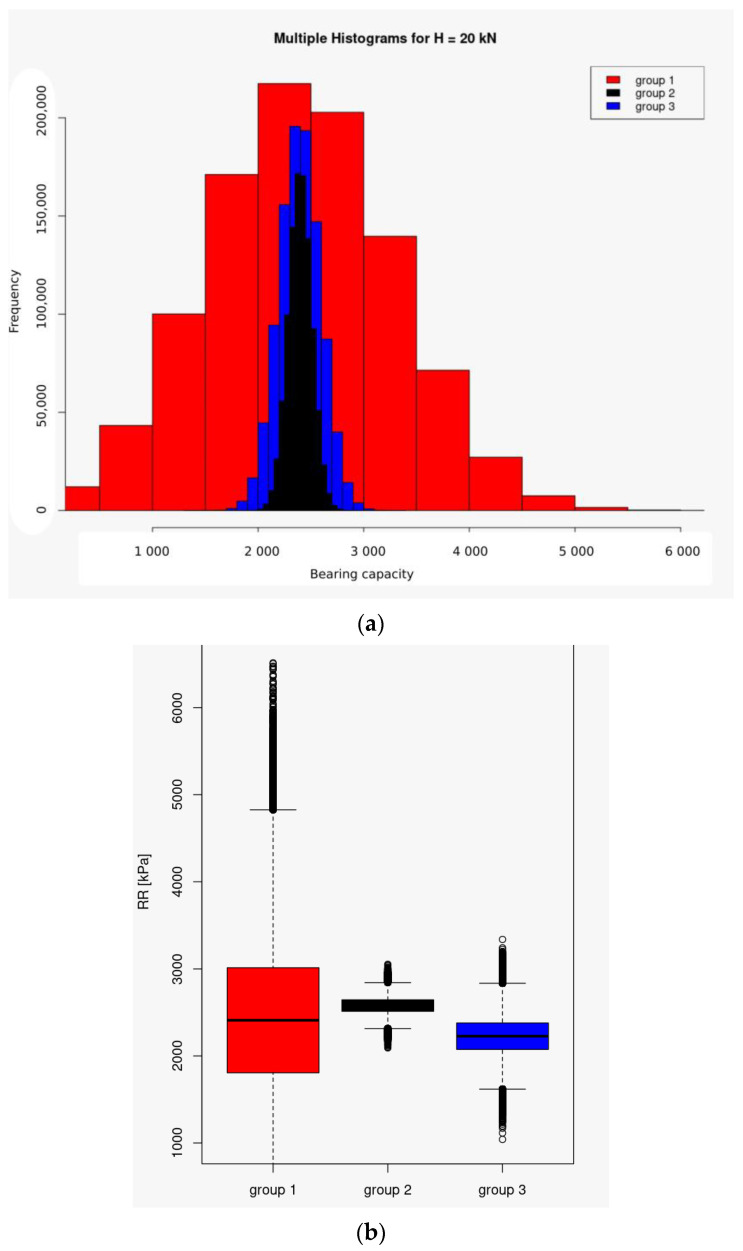
Comparison of bearing capacity values (on horizontal axes in kN) determined from testing only a selected group of sample sizes for: (**a**) multiple histograms for horizontal force of H = 20 kN; (**b**) basic statistical description for the tested groups in the form of box plots visualizing the Minimum Non-Outlier, First Quartile, Median, Third Quartile, and Maximum Non-Outlier for H = 20 kN.

**Table 1 materials-16-05631-t001:** Characteristics of the tested soil.

Properties	Standard	Value
Sand [%]	EN ISO 17892-4:2016 [46]	14
Silt [%]	EN ISO 17892-4:2016 [46]	63
Clay [%]	EN ISO 17892-4:2016 [46]	23
Liquid limit LL [%]	EN ISO 17892-12:2018 [47]	33.52
Plastic limit PL [%]	EN ISO 17892-12:2018 [47]	22.31
Optimum moisture content OMC [%]	EN 13286-2:2010 [48]	16.07
Maximum dry density MDD [t·m^−3^]	EN 13286-2:2010 [48]	1.788
Permeability coefficient k [m/s]	EN ISO 17892-11:2019 [49]	5.71 × 10^−10^
Specific surface area S_0_ [m^2^·g^−1^]	ISO 9277:2010 [50]	36.11

**Table 2 materials-16-05631-t002:** Comparison of test results for c_u_ for different sample dimensions.

Sample Number	Group 1 D = 38 mm 0.0867 dm^3^		Group 2 D = 70 mm 0.563 dm^3^		Group 3 D = 100 mm V = 1.6220 dm^3^	
	q_u_	c_u_	q_u_	c_u_	q_u_	c_u_
[−]	[kPa]	[kPa]	[kPa]	[kPa]	[kPa]	[kPa]
1	326.62	163.31	225.82	112.91	216.42	108.21
2	246.35	123.17	205.88	102.94	186.42	93.21
3	234.64	117.32	204.35	102.17	152.64	76.32
4	287.88	143.94	203.15	101.58	162.60	81.30
5	278.51	139.25	213.02	106.51	184.20	92.10
6	276.03	138.02	215.18	107.59	149.12	74.56
7	285.23	142.62	221.50	110.75	168.12	84.06
8	249.50	124.75	218.65	109.33	191.15	95.58
9	259.60	129.80	211.18	105.59	194.85	97.43
10	290.12	145.06	210.50	105.25	197.98	98.99
11	265.84	132.92	202.98	101.49	200.05	100.03
12	279.65	139.83	215.60	107.80	194.18	97.09
13	295.32	147.66	214.50	107.25	212.50	106.25
14	276.42	138.21	213.20	106.60	179.80	89.90
15	273.35	136.68	204.80	102.40	187.31	93.66

**Table 3 materials-16-05631-t003:** Results of statistical analysis for undrained shear strength values c_u_.

Group of Samples	Number of Results	Mean c_u_ Value	Min. c_u_ Value	Max. c_u_ Value	Standard Deviation SD	Lower Confidence Limit α = 0.05 α = 0.01	Upper Confidence Limit α = 0.05 α = 0.01	Coefficient of Variation CV
	szt.	kPa	kPa	kPa	kPa	kPa	kPa	%
1	14	138.1	234.6	326.6	9.89	132.4 130.2	143.8 146.1	7.16
2	15	107.2	101.0	113.7	4.03	104.9 104.1	109.4 110.3	3.75
3	12	100.4	86.6	113.3	8.06	95.5 93.5	105.2 107.2	8.02

**Table 4 materials-16-05631-t004:** Values of exponential curve fitting to the measurement data.

Variable	Estimate	Std. Error	*t* Value	Pr (>|t|)
c_0_ [kPa]	152.0079	4.6587	32.629	<2 × 10^16^ ***
m [−]	3.5963	0.8557	4.203	0.000149 ***
c_upl_ [kPa]	100.2066	2.2632	44.277	<2 × 10^16^ ***

Signif. code: ‘***’ −0.001. Residual standard error: 7.64 on 39 degrees of freedom. Achieved convergence tolerance: 4.522 × 10^−6^.

**Table 5 materials-16-05631-t005:** Parameter fitting values as a function of confidence level (based on the t-Student distribution).

	τ [−]	c_0_ [kPa]	m [−]	c_upl_
1	−3.1219722	146.72	2.113197	92.31
2	−2.6168932	147.69	2.332755	93.76
3	−2.1036509	148.56	2.554942	95.14
4	−1.5845574	149.39	2.784932	96.46
5	−1.0611353	150.20	3.028935	97.74
6	−0.5344654	151.06	3.294794	98.98
7	0.0000000	152.01	3.596334	100.21
8	0.5469267	153.15	3.954846	101.43
9	1.0988844	154.61	4.396491	102.65
10	1.6486570	156.61	4.970571	103.84
11	2.1963951	159.77	5.799556	105.01
12	2.7422837	166.28	7.287252	106.17

**Table 6 materials-16-05631-t006:** Results of the mean (E) and standard deviation (SD) for the input distributions and the result variable (11) and (12).

Sample Group	Point Estimation of the c_u_ Distribution [kPa]	Mean Volume [dm^3^] and Diameter [mm] for Group	Point Estimation of c_upl_ [kPa]
1	mean = 138.1, SD = 9.89	0.0867, 38	mean = 99.6, SD = 36.9
2	mean = 107.2, SD = 4.03	0.5630, 70	mean = 100.4, SD = 4.66
3	mean = 100.4, SD = 8.06	1.6220, 100	mean = 100.3, SD = 8.13

**Table 7 materials-16-05631-t007:** Load bearing capacities of fundament under different design conditions.

c_upl_ [kPa]	Group	Static Solution	Acceleration (p-Wave)			
		a = 0 cm/s^2^	a = 40 cm/s^2^	a = 60 cm/s^2^	a = 80 cm/s^2^	a = 100 cm/s^2^
				H_dyn_ [kN]		
		a = 0 cm/s^2^	15.6	23.4	31.2	40
		Load Bearing Capacity R_k_ [kN]				
138.1	1	3707	3604	3571	3554	3537
107.4	2	2931	2829	2797	2780	2763
100.4	3	2764	2662	2630	2614	2595

**Table 8 materials-16-05631-t008:** Basic statistical description for simulations of laid capacity results based for groups of laboratory tests (H = 20 kN).

Statistical Parameter	Group 1	Group 2	Group 3
Min.	184	2093	1041
1st Qu.	1807	2513	2075
Median	2410	2580	2227
Mean	2412	2580	2227
3rd Qu.	3015	2646	2379
Max.	6516	3054	3340
SD	894.3	8.0	225.3

**Table 9 materials-16-05631-t009:** Comparison of possible scaling measures with comments.

Criterion	Group 1	Group 2	Group 3	Comments (Advantages and Disadvantages)
Base * diameter D (mm)	38	70	100	+ Dimensions are used in description of triaxial tests, + Direct description of the propagation of decrease pore pressure by the length, + linear scale, − No correlation with maximum energy dissipation in the specimen,
Base * area A (mm^2^)	1134	3848	7853	+ Describes the influence of contact phenomena, + Directly correlated with the load area, + Quadratic scale, + The same as in diameter, but for constant shape of samples, − No correlation with maximum energy dissipation in the specimen,
Sample volume V (dm^3^)	0.0867	0.563	1.622	+ Describes the maximum dissipation of energy, + Qubic scale, − Lost the proportion and length of pore pressure destress, − No slenderness effect, − No cover direct physical parameters,
D/D_50_	3447	6432	9203	+ Used in previous literature studies, − Same as in previous points,
Sample weight (N)	1.549	9.947	29.01	+ Describes the maximum dissipation of energy, + Qubic scale, + Clear relation to material, − Lost the proportion and length of pore pressure destress, − No slenderness effect

* The base refers to the lower surface of the cylinder (sample).

## Data Availability

The majority of the data used in this study are available within the article. Additional data that are not included in the article can be provided upon request.

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
