# Peer review of "Assessing the Scale Effect on Bearing Capacity of Undrained Subsoil: Implications for Seismic Resilience of Shallow Foundations"

_materials, 2023, doi:10.3390/ma16165631_

Round 1

Reviewer 1 Report

This study aimed to examine how the size of cohesive soil samples affects their bearing capacity under undrained conditions. The study looked at the compression response of three different sample sizes and used statistical analysis to develop models for predicting the seismic resilience of shallow foundations. However, the reasons for the scale effect of cohesive soils still need to be clarified, which raises some questions.

One question concerns the deformation and mechanism in the cylindrical specimen pertaining to its size. The paper's photographs and text show that all specimen groups have similar failure modes under compression loading. Another question concerns the compaction method and moisture content, which are crucial in the samples' strength. It's important to know whether manual or vibratory shaker compaction was used and the experimental protocol for ensuring all specimens had similar moisture content.

The literature review overlooks the numerous studies related to the numerical modelling of cohesive soils to predict their strength. Using the extended Drucker-Prager model could provide valuable insights into the state of stress in undrained compression tests. The author should consider employing this model to explain the scale effect, even though the compression tests involved only a limited group of specimen geometries. The literature review should be updated with the work related to numerical modelling.

The article is well-written with no major concerns regarding language usage.

Reviewer 2 Report

I have reviewed this article about seismic resiliency of foundations. 

It seems rather scientific and well developed

The concept of this article is interesting because it is an issue not usually identified in the engineering problems of soil mechanics.

That said, I think that this article is indeed a research article with few shortcomings to be considered but one of these is the lack of relationship of the tests with laws of elasticity and fracture of materials and proposals for simulations that can be applied to different contexts and simulations. In this sense, perhaps I would amplify the simulations to more cases, like for instance the one they have presented in Taiwan. Especially because Poland is not known to be an especially seismic area.

The methodology is perceived as a laboratory routine testing.  I miss some more scientific introductions to the problem mentioning three-dimensional aspects.

There is finite element simulation available for soil mechanics but the authors do not seem to apply such useful methodology.

So, the development of the article is promising but it lacks insights for generalization.

The authors possess knowledge on this matter but they fail to convey their findings to a wider audience of potential readers outside their particular laboratory testing.

There graphs are mainly statistic but somewhat detached from engineering practice, of building and infrastructures foundations.

The authors should perform a more significant effort to bring forth their valid discoveries.

Due to the former problems, the outcome of the manuscript is not entirely conclusive.

Summary of evaluation: This article is promising and well researched in the laboratory part. But some minor corrections and further developments are required by my part.

Reviewer 3 Report

The article is devoted to an important and topical issue related to the study of soil and the assessment of its impact on the stability of foundations.

The development of mineral deposits has a huge transformative effect on the geological environment by disturbing the relief, structure, condition, properties, temperature regime of rock masses. The scale of these impacts depends on the volume of extracted rocks, the depths of workings, and the technology of work. Thus, the introduction of man into the geological environment violates the natural balance that has developed in it, which leads to the development of engineering-geological processes in the sides of quarries, the walls of underground mine workings.

The reason for this is the transformation of fields: natural stresses, temperatures, and others. The formation of engineering-geological processes complicates the mining process and affects not only the safety of work, but also the economics of the enterprise. The study and analysis of the construction and operation of mine workings shows that the development of engineering-geological processes at deposits, especially those located in areas of permafrost development, arise due to incomplete consideration of individual features of individual sections of deposits in the design schemes.

In this case, special attention should be paid to the scale effect, which is not given enough attention in the literature.

Based on the foregoing, the presented work is relevant and will be of interest to the reader in the field under consideration.

However, a number of comments about the article should be clarified:

1. In the introduction, the literature review should be expanded by a more detailed consideration of the scale effect in relation to metals and various materials made of metals. In particular, the following works could be considered: https://doi.org/10.1016/j.ijplas.2022.103406, https://doi.org/10.3390/ma16093490, https://doi.org/10.1016/j.jmps.2023.105222.

2. More references for the last 5 years should be added to the list of references.

3. It is necessary to characterize the circuit shown in Figure 1 in more detail. And also explain what advantages the use of the presented circuit has over known analogues.

4. Has a comparative analysis of the considered fine-grained soil in Poland (Figure 2) been carried out with similar soils in other regions of the world? Is it possible to apply the developed technique to other soils?

5. How were the main soil and moisture parameters determined (Table 1)? Was the method of expert assessments, cluster analysis, etc. used to select the most important parameters?

6. Section "2.2 Laboratory Tests" should explain in more detail what is meant by a horizontal seismic wave. Perhaps practical examples should be given.

7. Based on the dependences shown in Figure 6, a regression analysis should be carried out to obtain the corresponding mathematical models.

8. The numerous practical results obtained in the work should be given in the conclusions more fully.

9. In the conclusions, one could dwell in more detail on the approbation of the results obtained on specific objects and the prospects for further research in other regions of the world.

Reviewer 4 Report

The objective of this paper is to assess the scale effect on bearing capacity of undrained subsoil and particularly the implications for seismic resilience of shallow foundations. Moreover, this paper analyzes the influence of the scale effect on the strength of soil under unidirectional stress conditions during undrained shear (cu), while the impact of this parameter on the bearing capacity of shallow foundations during seismic actions is investigated. Finally, an analysis was conducted to minimize the effects of the scale effect and estimate the level of safety depending on the applied standardized measurement method.

This is an interesting and well-structured paper. All necessary sections (Introduction, Materials and Methods, Results, Discussion, Conclusions) have been considered. Moreover, the “Materials and Methods” section is divided into sub-sections, providing additional details. Furthermore, all Figures, Tables and Diagrams are consistent with the analysis provided in the manuscript. Regarding the mathematical part, predominantly analyzed in the “Materials and Methods” and “Results” sections, it is valid and satisfactorily explained. However, some changes should be implemented, which will improve the paper. In particular:

Lines 103: The workflow in Figure 1 is confusing, as it includes a lot of information (many images and unnecessary text). I suggest splitting it up into two Figures, while the text can be included in the captions. Moreover, the Figures should be provided in a higher resolution, as many blurry parts are included in the current form. Please, apply.

Line 153: Before analyzing the seismic effect on buildings, I suggest adding a brief paragraph, explaining the earthquake phenomenon and how seismic waves occur. Typical papers, in which the corresponding information can be obtained and optionally be cited, are the following: 1. Mildon, Z. K., Toda, S., Faure Walker, J. P., & Roberts, G. P. (2016). Evaluating models of Coulomb stress transfer: Is variable fault geometry important? Geophysical Research Letters, 43(24). https://doi.org/10.1002/2016GL071128, 2. Sboras, S., Lazos, I., Bitharis, S., Pikridas, C., Galanakis, D., Fotiou, A., Chatzipetros, A., Pavlides, S., 2021. Source modelling and stress transfer scenarios of the October 30, 2020 Samos earthquake: seismotectonic implications. Turkish Journal of Earth Sciences 30, 699–717. https://doi.org/10.3906/yer-2107-25, 3. Toda, S., & Enescu, B. (2011). Rate/state Coulomb stress transfer model for the CSEP Japan seismicity forecast. Earth, Planets and Space, 63(3), 171–185. https://doi.org/10.5047/eps.2011.01.004. Please, apply.

Lines 214 and 222: Please, provide a more detailed description in the Figure 5 and Figure 6 captions, respectively.

Lines 257 and 261: Please, provide a more detailed description in the Figure 8 and Figure 9 captions, respectively.

Lines 336 and 339: Please, provide Figure 12 and Figure 13 in a higher resolution, respectively.

Line 432: The “Conclusions” section should be modified. In its current form, it resembles an extended abstract rather than conclusions. This section should be comprehensive, while the major findings of the paper should be highlighted. Maybe, numbering of the concluding remarks could be performed. Please, apply.

Round 2

Reviewer 3 Report

The authors made the necessary changes to the article. I recommend the article for publication in its present form.